# Constrained Human-AI Cooperation: An Inclusive Embodied Social Intelligence Challenge

**Weihua Du**[*1♭]**, Qiushi Lyu**[*2♯]**, Jiaming Shan**[3]**, Zhenting Qi**[4]**, Hongxin Zhang**[5]**, Sunli Chen**[5]
**Andi Peng**[6]**, Tianmin Shu**[7]**, Kwonjoon Lee**[8]**, Behzad Dariush**[8]**, Chuang Gan**[5♩]

[1] Carnegie Mellon University, [2] Peking University, [3] University of California, Santa Barbara
[4] Harvard University, [5] University of Massachusetts Amherst, [6] MIT
[7] Johns Hopkins University, [8] Honda Research Institute USA

[♭] `weihuad@cs.cmu.edu`, [♯] `lvqiushi@stu.pku.edu.cn`, [♩] `chuangg@umass.edu`

## Abstract

We introduce Constrained Human-AI Cooperation (CHAIC), an inclusive embodied social intelligence challenge designed to test social perception and cooperation in embodied agents. In CHAIC, the goal is for an embodied agent equipped with egocentric observations to assist a human who may be operating under physical constraints—e.g., unable to reach high places or confined to a wheelchair—in performing common household or outdoor tasks as efficiently as possible. To achieve this, a successful helper must: (1) infer the human's intents and constraints by following the human and observing their behaviors (social perception), and (2) make a cooperative plan tailored to the human partner to solve the task as quickly as possible, working together as a team (cooperative planning). To benchmark this challenge, we create four new agents with real physical constraints and eight long-horizon tasks featuring both indoor and outdoor scenes with various constraints, emergency events, and potential risks. We benchmark planning- and learning-based baselines on the challenge and introduce a new method that leverages large language models and behavior modeling. Empirical evaluations demonstrate the effectiveness of our benchmark in enabling systematic assessment of key aspects of machine social intelligence. Our benchmark and code are publicly available at `https://github.com/UMass-Foundation-Model/CHAIC`.

## 1 Introduction

Humans possess a remarkable ability to observe, infer, and help others, even when others have different mental models and physical constraints in the world from themselves (Warneken and Tomasello, 2006). From a young age, humans are able to watch other people attempt to perform a task, and if other people fail, they can develop plans of action that best assist them. In contrast, AI agents struggle to exhibit such basic social skills and fail to adjust their plans for the specific humans they wish to aid (Valmeekam et al., 2022; Ngo et al., 2022), rendering them poor personalized helpers.

For AI agents to best assist human partners in performing tasks in the real world, they must possess two fundamental capabilities: (1) contextual perception, i.e., the ability to follow and observe human behavior and identify the specific goals and constraints faced by each human; and (2) cooperative planning, i.e., the ability to plan actions that are best tailored to helping each human with different goals and constraints. While there have been some embodied benchmarks and environments designed to test general multi-agent intelligence (Puig et al., 2021, 2023b; Gan et al., 2021), such efforts have largely excluded the unique accessibility challenges that real humans may possess in the world and

---

[*]Equal Contribution

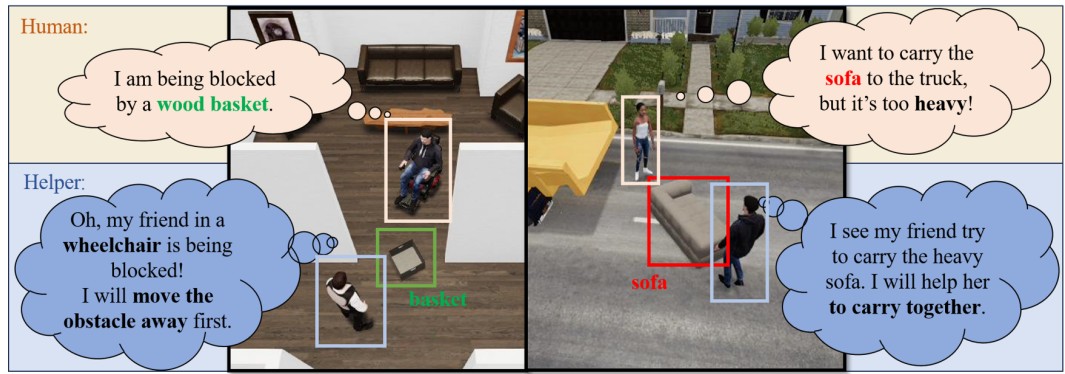

Figure 1: **Constrained Human-AI Cooperation (CHAIC) for benchmarking embodied agents that socially perceive and assist human partners with physical constraints.** Left: A human partner is confined to a wheelchair and struggles to move past an obstacle. The helper agent infers the human partner's constraints and intents and assists him by removing the obstacle. Right: In a moving house scenario, after observing a human partner fail to lift heavy furniture, the helper agent understands her intents and constraints and assists her in carrying the furniture together.

neglect the differences among individuals. Moreover, outdoor scenarios and emergencies are also prevalent in human life, but receive little attention in the embodied intelligence community (Deitke et al., 2022).

This paper introduces the first large-scale embodied social intelligence challenge with accessibility explicitly in mind: Constrained Human-AI Cooperation (CHAIC). In this challenge, an embodied agent with egocentric visual observation must actively perceive and cooperate with a human partner possibly with physical constraints in a near photo- and physically realistic virtual environment to complete common household and outdoor tasks as efficiently as possible. This is motivated by the idea that people who need the most help from autonomous agents are those who are currently not explicitly accounted for in embodied intelligence frameworks. In CHAIC, a helper agent needs to follow and *observe* the human partner to *infer* their goals and constraints; then, the agent *plans* a user-tailored strategy for aiding the human in efficiently performing tasks together; moreover, with the existence of unexpected emergencies, the agent needs to be reactive and adjust its strategy accordingly.

To create the challenge with accessibility in mind, we design and implement four new agents with real physical constraints that reflect the rich diversity of human partners in the real world. For example, a human partner confined to a wheelchair struggles to move past obstacles or a human partner struggles with heavy furniture when moving house in an outdoor scene, shown in Figure 1, and eight long-horizon tasks featuring both indoor and outdoor scenes on top of the ThreeDWorld (Gan et al., 2021), explicitly motivating the development of embodied agents that prioritize accessibility efforts when learning and planning and can thrive in rich scenarios.

We benchmark several baseline models, including planning- and learning-based agents, especially those powered by foundation models. We also introduce a new method for building agents that combines the behavior modeling capabilities of video models with the reasoning ability of large language models. Our benchmark results suggest that current baselines have difficulty modeling partner behaviors from raw RGB images, and LLM-driven agents are competitive agents in decision-making. We hope this new challenge will advance the study of social intelligence in embodied agents in complex scenarios including diverse human partners with constraints and rich indoor and outdoor scenes. This initiative calls on the community to develop and evaluate embodied agents with a strong emphasis on accessibility and inclusivity.

Our contributions include:

- We design and implement four new agents with real physical constraints and eight long-horizon tasks featuring both indoor and outdoor scenes on top of ThreeDWorld (Gan et al., 2021), simulating rich human constraints and scenarios in the real world.

- We introduce a new embodied social intelligence challenge with accessibility explicitly in mind: Constrained Human-AI Cooperation (CHAIC), to test embodied agents' ability to actively perceive human partners' intents and constraints from egocentric visual observations and make user-tailored cooperative plans to help constrained human partners in rich scenarios.

- We benchmark several baseline models, including those powered by foundation models, especially a new agent with behavior modeling introduced by us, and conduct comprehensive analyses to identify and discuss the persisting challenges related to inter-agent perception and cooperation within complex environments.

## 2 Related Work

**Embodied Multi-Agent Cooperation Challenges**   Our benchmark and environment build on a rich history of realistic 3D simulated environments (Zhou et al., 2024; Li et al., 2023; Padmakumar et al., 2022; Kolve et al., 2017; Shridhar et al., 2020; Misra et al., 2018; Zhu et al., 2017; Xia et al., 2018; Savva et al., 2019; Xiang et al., 2020). Various tasks and methods have been introduced for multi-agent cooperation (Lowe et al., 2017; Samvelyan et al., 2019; Carroll et al., 2019; Suarez et al., 2019; Jaderberg et al., 2019; Amato et al., 2019; Baker et al., 2020; Bard et al., 2020; Jain et al., 2020; Puig et al., 2023b; Wen et al., 2022; Szot et al., 2023; Zhang et al., 2023, 2024). Specifically, Puig et al. (2021, 2023a) explored the inter-agent perception of isomorphic agents during household tasks. However, these works did not address the explicit challenge of actively perceiving diverse human partners with physical constraints from visual observations and adapting the cooperation strategy accordingly. In contrast, our challenge is designed explicitly not only to study the social perception of the partner's goals and constraints from visual observations but also to capture the nuances of human physical mobility constraints that might impair the successful completion of such tasks. A contemporary work (Cao et al., 2024) also studies assistive agents for vulnerable groups but focuses only on indoor scenarios with oracle symbolic observations. In contrast, our proposed CHAIC Challenge features both indoor and outdoor scenarios, an egocentric visual observation, newly created physically constrained agents, and unexpected events, enabling rich, physics-driven interactions on real-world assistive tasks.

**Accessibility in AI Design**   People with disabilities or physical impairments are a central focus area of study in robotics, including care for wheelchair users, elderly users, and users with aging-related ailments like dementia (de Saille et al., 2022; Sundaresan et al., 2022; Broadbent et al., 2009; Benda et al., 2020; Cooper et al., 2016; Lee et al., 2017). These works often study the best ways to design for inclusivity; in other words, how to best build assistive robots to handle the *explicit* physical needs of the users in question (Benda et al., 2020). We build on these design principles to create the first-ever large-scale embodied intelligence environment that explicitly models such impairments.

## 3 The Constrained Human-AI Cooperation (CHAIC) Challenge

**The Constrained Human-AI Cooperation (CHAIC) Challenge** seeks to study how embodied agents perform in terms of social perceptions of human partners with diverse physical constraints and cooperative planning abilities within rich scenarios. Built on top of ThreeDWorld, a realistic 3D embodied AI platform, we design and implement four new agents with real physical constraints (Section 3.1) and eight tasks featuring both indoor and outdoor scenes, including emergencies (Section 3.2). For each task, there is a **constrained agent** mimicking a human partner with capability constraints, trying to find and transport some target objects to a specific goal location, and a **helper agent** trying to infer the constrained agent's goal and capability constraints through active perception of its behaviors to assist the constrained agent better. The success of the helper agent is measured by the ratio of target objects successfully transported by both of them. Figure 2 provides an overview of the challenge, with further details in Section 3.3.

### 3.1 Constrained Agents

To enable the testing of embodied social intelligence with a diverse set of potential human partners, we have created four new simulated agents that may face physical constraints such as limited height, strength, and movement speed, reflective of real humans.

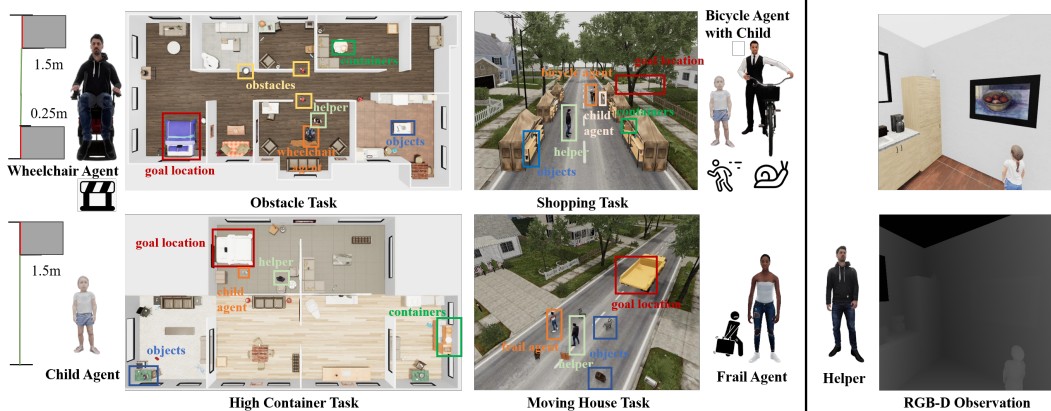

Figure 2: **Overview of CHAIC challenge:** We present four agents with diverse capability constraints and eight tasks built around these constraints, featuring both indoor and outdoor scenarios. The tasks are named *no constraint*, *high container*, *high goal location*, *high target object*, *obstacle*, *low target object*, *shopping*, and *moving house* (four of them are shown on the left). In each task, there are *objects*, *containers*, and a *goal location*. A helper agent needs to infer the partner's intents and constraints from its egocentric observations (shown on the right) and make a tailored plan to assist the partner in transporting the intended target objects to the goal location using containers as tools.

Each agent possesses two properties: *reaching range* and *strength*. An agent can successfully interact with objects whose heights are within its *reaching range* and whose weights are lighter than its *strength* limit. When an agent attempts an action that exceeds its capabilities, the action does not fail immediately but instead has a success rate. This rate is calculated using the formula $\exp(-\delta/\alpha)/\beta$, where $\delta$ represents the excess amount, and $\alpha$ and $\beta$ are constants. If an action exceeds multiple capability thresholds, the probabilities of success are multiplied. We have developed the following constrained agents:

- **Child Agent:** A small child with a height of 1.2 m that has a *reaching range* of $[0, 1.5]$ m.

- **Wheelchair Agent:** An agent confined to a wheelchair or limping that may be blocked by obstacles in the house (e.g., a couch). Its *reaching range* is $[0.25, 1.5]$ m.

- **Bicycle Agent:** An agent walking with a bike that moves slowly. It must first dock the bike when picking up an object. The child accompanying it may run away, causing an emergency.

- **Frail Agent:** An agent that is less capable of lifting heavy objects (e.g., furniture) and has only $1/6$ the *strength* of a normal agent.

### 3.2 Tasks with Constrained Agents

We designed eight tasks featuring indoor and outdoor scenes, including emergencies, in our CHAIC benchmark, utilizing the various constrained agents introduced earlier. Information about each task is shown in Table 1.

### 3.3 Challenge Details

In CHAIC, an embodied helper agent $A_h$ is tasked to infer the goal $G$ and the constraints of a constrained agent $A_m$ and assist $A_m$ in finding and transporting a set of target objects $O_t$ from random locations to a goal location $L_g$. There are containers scattered in the environment, which the agents can use to transport more objects simultaneously. An agent could take two objects at a time without a container, and the capacity of a container is set to three.

Formally, a task in the challenge is defined by the goal $G$ of the constrained agent $A_m$ (i.e., a set of goal predicates describing the final desired state) and an initial environment $E$ where the helper agent $A_h$ is placed alongside the constrained agent $A_m$ to complete the task. The ground truth goals and constraints of the constrained agent are hidden from the helper agent $A_h$, thereby explicitly motivating the need for active perception for the agent to infer intents and constraints.

Table 1: Tasks with constrained agents, including both indoor and outdoor scenes and rich features.

| Task Name | Scene | Description | Agent Type | Features |
|---|---|---|---|---|
| No constraint | Indoor | Main agent with no constraints | Normal agent | N/A |
| Low target | Indoor | Target objects on the ground | Wheelchair agent | N/A |
| Obstacle | Indoor | Obstacles between most rooms | Wheelchair agent | Existence of obstacles |
| High target | Indoor | Target objects in high places | Child agent | Fragile high targets may break |
| High goal location | Indoor | Goal locations in high places | Child agent | Fragile high targets may break |
| High container | Indoor | Containers in high places | Child agent | Fragile high targets may break |
| Shopping | Outdoor | Main agent walks a bike with his child while shopping | Bicycle agent | Emergency event: the child runs away |
| Moving house | Outdoor | Main agent moves all the furniture onto the truck | Frail agent | Agents can cooperate to lift furniture together |

**Observation Space**   In CHAIC, actions may take several frames to finish and are executed asynchronously between agents. The agent will receive the following observation after its action is finished or failed:

- **Egocentric RGB-D Image**: The agent receives $512 \times 512$ egocentric color and depth images, as shown in Figure 2.

- **Self-State**: The agent is provided with information relevant to itself, including its current location, orientation, and the objects in its possession.

**Action Space**   The action space consists of three low-level navigation actions (*move forward*, *turn left*, *turn right*), three basic interaction actions (*pick up A*, *put A in B*, *put A on B*), and one idle action (*wait*).

### 3.3.1   Task Generation

**Indoor Task**   To generate an indoor task, a floorplan configuration with six to eight interconnected rooms and a target task is initially sampled from predefined sets. For each scene, objects related to goals in the predefined set are placed on low surfaces such as tables, chairs, sofas, and floors for low objects, and higher surfaces like cabinets or refrigerators for high objects. However, only a subset of the objects is the target object set. The target object set is a set that includes all objects related to a specific type like *food* or *fruit*, one object randomly selected from a non-target set, and two additional fragile vases if the task is a high-target task. The number of targets is around ten. Then, a goal location and up to six containers are added to the scene based on available space and task constraints.

We randomly initialize two agents (one constrained agent $A_m$ and one helper $A_h$), and each agent is placed in a free space at least 0.5 meters away from the nearest wall. This setup ensures sufficient initial distance between the agents and the walls, allowing unrestricted movement at the beginning of the task.

**Outdoor Task**   The generation of outdoor tasks is largely the same as the indoor task generation. For the shopping task, six shops are generated and spread out on both sides of the road, and each shop sells one specific category of items. The goal location of the shopping task is a fixed, predetermined place in front of the bicycle agent's house. In the moving house task, the target objects include five pieces of furniture on the road in front of a house. The goal location is a truck parked nearby. The details of outdoor task generation can be found in Appendix G.

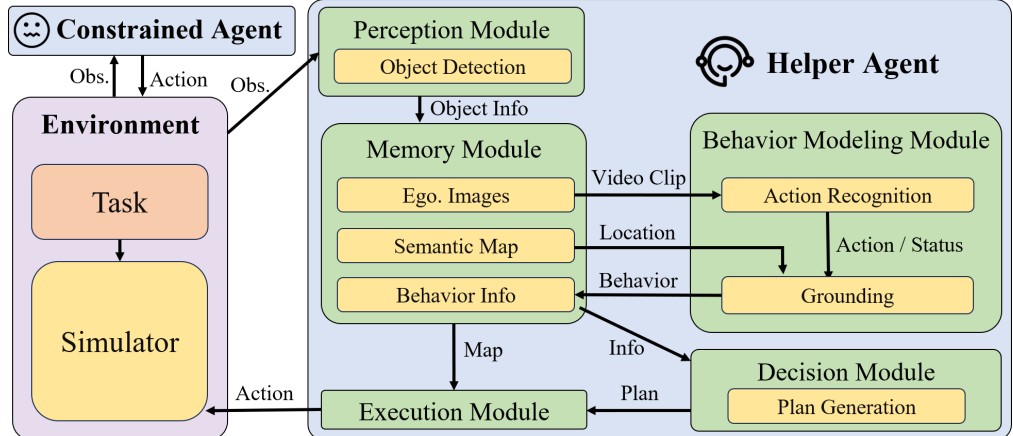

Figure 3: **LLM+BM Helper Implementation Pipeline:** An overview of the LLM+BM Helper with specific modules for *Perception*, *Behavior Modeling*, *Decision*, and *Execution*. (1) The **perception module** detects objects from raw RGB images; (2) the **memory module** builds the semantic map of the environment using depth images and records behaviors; (3) the **behavior modeling module** recognizes the action of the partner and localizes the object corresponding to the action; (4) the **decision module** decides plans for the next steps using foundation models; and (5) the **execution module** generates low-level actions.

### 3.3.2 Dataset Construction

For each of the eight tasks, we create 12 episodes for training and 12 episodes for testing, resulting in approximately 200 episodes in total. We ensure that the environments of the test set are different from those of the training set. We randomly sample the initial starting states for each task. An episode terminates when all goal predicates of the task are satisfied or when the maximum time step horizon $T = 3000$ frames is reached (for the moving furniture task, the maximum time step horizon is $T = 1500$).

## 4 Language Agent Augmented with Behavior Modeling Module

We also introduce a new agent framework combining the prowess of action recognition models and the reasoning ability of large language models (LLMs). Due to their simplicity and generalization ability, LLMs can also be implemented in other environments or the real world. We built a **behavior modeling module**, which models the behaviors of the constrained agent via an action recognition model and incorporated it into the CoELA framework (Zhang et al., 2023) with four other modules: (1) the **perception module**, which transforms the raw RGB-D observations into structured semantic maps via an object detection model; (2) the **memory module**, which saves all the history information in a structured manner; (3) the **decision module**, which generates high-level plans and is driven by large language models; and (4) the **execution module**, which turns the generated plans into low-level actions. More details regarding these modules can be found in Appendix B.1. Figure 3 shows an overview of the framework.

### 4.1 Behavior Modeling Module

To infer the intents and inabilities of constrained agents, the behavior modeling module extracts constrained agents' actions and status from a sequence of egocentric images (i.e., a video). The behavior modeling module contains two parts: **action recognition** and **action grounding**.

**Action Recognition** We adopt an action recognition model to enable the helper agent to recognize the actions of the constrained agent. We select the TSN model (Wang et al., 2016) pretrained on Kinetics-400 (Kay et al., 2017) as the base video action recognition model. There are four types of actions: *pick up*, *put on*, *put in*, and *walking* (including *move forward*, *turn left*, and *turn right*). Each action may be successfully executed or fail (except for *put in* and *walking*, which are always

successful), so there are six classes in total. We collect data by having an agent follow the constrained agent to observe its behaviors while executing a task and store the action video clips in the training set. During testing, the helper agent utilizes this model to recognize the actions of the constrained agent when it is within observation. The training details can be found in Appendix D.2.

**Action Grounding**    After the helper recognizes the action of the constrained agent, it looks up the semantic map in the memory module for the predicate of the action. For example, when the action is *pick up*, the predicate will be the nearest object to the constrained agent. Finally, the behavior modeling module identifies the action, the predicate of the action, and the status of the action of the constrained agent.

## 5    Experiments

### 5.1    Setup

#### 5.1.1    Constrained Agent Policy

The constrained agent takes ground truth object segmentation as observation to mitigate the impact of imperfect visual perception on performance and chooses actions based on a rule-based high-level planner designed with handwritten rules by human experts.

At the beginning of an episode, the constrained agent will *explore* the environment to find more target objects, containers, and the goal location. Whenever the agent finds target objects or containers, it will pick them up if it has free hands. If more than $50\%$ of the time steps are left and it does not have a container in hand, its priority will be to *pick up a container*; otherwise, it will *pick up a target*. If the agent cannot carry more objects, it will *put the object on the goal location*. If less than $25\%$ of the time steps is left ($37.5\%$ if it has not found the goal location yet) and the agent is carrying a target object, it will *put the object on the goal location* immediately since it is often a long walk to the goal location. When selecting possible targets, the agent will opt for the closest one if multiple options are available. Moreover, at any time, if the agent can *put an object in a container*, it will do so.

#### 5.1.2    Evaluation Metrics

To evaluate the success of helper agents, we measure the following three metrics:

- **Transport rate (TR):** The percentage of target objects that the agents successfully transported. We also calculate the **Efficiency Improvement (EI)** of having the helper as $\Delta M/M_0$, where $\Delta M$ denotes the increase in the transport rate after adding the helper, and $M_0$ denotes the larger of the transport rates of the team or the constrained agent alone, for numerical stability.

- **Goal Inference Accuracy (IA):** The ratio of target objects successfully transported by the helper to the total number of objects transported by the helper.

- **Emergency Rate (ER):** For the shopping task, we calculate the ratio of frames where the child agent is away from the constrained agent to measure the helper agent's ability to handle emergencies.

### 5.2    Baselines

We test four types of planning-based helpers: Random Helper, Rule-Based Hierarchical Plan Helper (RHP), LLM+BM Helper, and VLM Helper. All the helpers share the same *Perception Module*, *Memory Module*, and *Execution Module* as the language agent introduced in Section 4, but the critical differences lie in the high-level planner. Meanwhile, an Oracle Helper is tested to demonstrate the upper-bound performance. Below is the description of each type of helper:

- **No Helper (w/o):** The constrained agent performs the task solely without assistance from a helper.

- **Random Helper:** A naive helper randomly selects a plan from a list of valid plans.

- **Rule-Based Hierarchical Plan Helper (RHP):** This helper uses prior knowledge of the task and relies on handcrafted rules by human experts to make plans to assist the constrained agent in completing the task. Further details on the rules can be found in Appendix B.2.

Table 2: **Quantitative results on CHAIC benchmark.** We report the average *Transport Rate (TR)*, *Efficiency Improvement (EI)* and *Goal Inference Accuracy (IA)* here. **w/o** means the main agent does the task solely without a helper. The *Emergency Rate (ER)* metric is also reported for the shopping task.

| | Indoor | | | | | | | |
| --- | --- | --- | --- | --- | --- | --- | --- | --- |
| Helper Agent | No Constraint | | High Target | | High Container | | High Goalplace | |
| | TR(EI)↑ | IA↑ | TR(EI)↑ | IA↑ | TR(EI)↑ | IA↑ | TR(EI)↑ | IA↑ |
| w/o | 0.53 | / | 0.30 | / | 0.37 | / | 0.28 | / |
| Random | 0.52(-0.02) | 0.24 | 0.27(-0.05) | 0.29 | 0.36(0.00) | 0.25 | 0.33(0.10) | 0.14 |
| RHP | 0.64(0.15) | 0.15 | 0.35(0.11) | 0.29 | 0.45(0.19) | 0.21 | 0.35(0.18) | 0.21 |
| VLM (GPT-4o) | 0.63(0.14) | 0.24 | 0.33(0.06) | **0.32** | 0.43(0.12) | **0.40** | 0.26(-0.20) | 0.33 |
| LLM (GPT-4) + BM | **0.65(0.17)** | **0.25** | **0.38(0.19)** | 0.29 | **0.49(0.24)** | 0.30 | **0.36(0.23)** | **0.35** |
| Oracle | 0.77(0.31) | 0.88 | 0.49(0.37) | 0.91 | 0.69(0.47) | 0.91 | 0.61(0.56) | 0.90 |

| | Indoor | | | | Outdoor | | | |
| --- | --- | --- | --- | --- | --- | --- | --- | --- |
| Helper Agent | Low Target | | Obstacle | | Shopping | | | Furniture |
| | TR(EI)↑ | IA↑ | TR(EI)↑ | IA↑ | TR(EI)↑ | IA↑ | ER↓ | TR(EI)↑ |
| w/o | 0.51 | / | 0.07 | / | 0.37 | / | / | 0.17 |
| Random | 0.50(-0.01) | 0.31 | 0.21(0.56) | 0.24 | 0.39(0.05) | 0.34 | 0.32 | 0.48(0.68) |
| RHP | 0.66(0.23) | 0.28 | **0.44**(0.77) | 0.17 | 0.49(0.22) | 0.44 | **0.30** | 0.65(0.72) |
| VLM (GPT-4o) | 0.69(0.26) | **0.46** | 0.40(0.86) | 0.35 | 0.50(0.25) | 0.72 | 0.39 | **0.70(0.78)** |
| LLM (GPT-4) + BM | **0.70(0.27)** | 0.43 | 0.42(**0.89**) | **0.47** | **0.58(0.33)** | **0.74** | 0.38 | 0.69(0.77) |
| Oracle | 0.82(0.38) | 0.91 | 0.60(0.87) | 0.82 | 0.61(0.39) | 0.87 | 0.17 | 0.76(0.80) |

- **LLM+BM Helper:** A language agent augmented with a Behavior Modeling module introduced in Section 4. Example prompts can be found in Appendix C.1. We use GPT-4 as our decision-making LLM.

- **VLM Helper:** A vision-language agent similar to the LLM+BM Helper. The last 10 frames of egocentric RGB-D observation are added as visual inputs to perceive the constrained agent. We use GPT-4o as our decision-making VLM.[2]

- **Oracle Helper:** An oracle helper that knows the ground truth goal, as the ground truth object segmentation, and the task progress. It behaves the same way as the RHP and is close to the upper-bound performance a helper could achieve.

We also tested some learning-based methods like reinforcement learning and SmartHelp (Cao et al., 2024), whose results can be found in Appendix E.3.

### 5.3 Main Results

We conducted an extensive evaluation by deploying four baseline models across eight distinct constraint settings and measured four specific metrics, as outlined in Section 5.1.2. The results are presented in Table 2. Overall, the LLM+BM Helper emerges as a strong baseline, achieving the highest transport rate (TR) in 6 out of 8 tasks, the most significant efficiency improvement (EI) in 7 out of 8 tasks, and the best goal inference accuracy (IA) in 4 out of 8 tasks.

**Behavior Modeling Analysis** Our LLM+BM Helper achieves a reasonable IA metric compared with other helpers, which shows our behavior model successfully models the partner's behaviors to some extent. However, compared with the Oracle Helper, all the other baseline agents perform poorly on the IA metric. The IA metric reflects whether the helper successfully determines the needs of the constrained agent, so the gap shows all our baselines do not work well in inferring the behavior of the constrained agent from the raw RGB-D image sequence. Nevertheless, our fine-tuned action recognition model achieves $86\%$ accuracy on the validation set (See Appendix D.2 for the action recognition model details). Two reasons contribute to the discrepancy: (1) Due to blocking or distance, the action clip received by the helper may be incomplete or out-of-distribution from training data. (2) The current LLM-based decision module is insufficient to balance observing the partner's behavior and acting independently.

---

[2]The main experiments are carried out between May 28 to Jun.5, 2024.

**LLM Can Infer Goals Correctly and Perform Actions Properly**   In analyzing some of the chain-of-thought outputs of LLM, we observe that the LLM-based helper can accurately infer the target objects desired by the constrained agent and formulate appropriate plans to collect them. For instance, in an outdoor shopping scene, the bike agent named David seeks some fruit. Initially, the LLM helper assesses, *"Since David hasn't picked any object yet, it's challenging to precisely determine his target objects."* It then realizes, *"No matter what object David wants, the best first step would be to maximize the efficiency of carrying objects by using a container,"* and subsequently proceeds to pick up a container. Upon observing the bike agent picking an apple, the LLM helper deduces, *"Considering the constraints and the objects David has shown interest in (i.e., an apple), the best course of action from the provided list would be to 'goto and pick up target <apple>'."* With a container and a target object in both hands, the LLM helper notes, *"Considering I am currently holding two target objects (one directly and one in a container), the optimal next action is to put the object in your one hand to the container in your other hand. This action will free up one of my hands, allowing me to pick up more target objects and transport them efficiently to the goal."*

Meanwhile, the LLM helper is capable of picking other fruits besides apples, demonstrating its accuracy in inferring the object category. After freeing up one hand, the LLM helper states, *"Based on the observed actions and status of David, it's clear that his target objects are fruits, specifically apples...so picking up more grapes aligns with the goal."* Finally, after collecting several fruits and having both hands full, the LLM helper concludes, *"the best action to take next is to 'transport object in hand to goal space'. This action involves taking the container filled with target objects, along with the additional grape in the other hand, to the specified goal location."* The detailed analysis of these chain-of-thought outputs is shown in Appendix F.1.

**Dealing with Emergencies**   In outdoor shopping tasks, the helper needs to handle unpredicted emergencies, requiring swift responses. The Emergency Rate (ER) metric shows that even if LLM- and VLM-based helpers can achieve high scores in normal tasks, they cannot handle emergencies as efficiently as RHP. To improve, some rule-based control may be required in LLM- and VLM-based helpers to help them prioritize and respond more effectively in urgent situations.

**Failure Case Analysis**   During the experiment, we discovered some common failure situations leading to poor performance, which might be helpful for further helper design.

- **Spatial Information Analysis**: The LLM-based agents do not understand spatial information very well when provided with text inputs of object locations. They often choose a distant object rather than a nearby one, even if they share the same name. Additionally, they often underestimate the cost of reaching the goal location and fail to transport due to time limits.

- **Acting without Cooperation**: In the obstacle task, a reasonable solution for the helper is to remove obstacles first to free the constrained agent. However, LLM- and VLM-based helpers often transport objects alone without assisting the constrained agent, leading to relatively bad performance in this task.

- **VLM is Unable to Infer the Targets Needed by Constrained Agents**: In certain tasks, the VLM Helper baseline performs worse than both the random baseline and the No Helper baseline. This is primarily because VLM cannot accurately infer the preferred objects of constrained agents when they observe them picking up items, leading it to consistently follow. Consequently, the VLM Helper fails to transport any objects, making it less effective than randomly transporting some objects, as done by the random helper. Additionally, frequently following the constrained agent can interfere with their actions—such as blocking their path—resulting in the VLM Helper baseline sometimes performing worse than having no helper. However, the LLM+BM Helper transports some objects even if it does not infer the goal correctly from the BM model, achieving a relatively higher score than the VLM Helper.

## 6   Conclusion

In this work, we proposed an accessibility-centered embodied social intelligence challenge: the Constrained Human-AI Cooperation (CHAIC) Challenge. This challenge includes four new agents with physical constraints and eight long-horizon tasks featuring both indoor and outdoor scenes, designed to test the critical skills of social perception and cooperation in embodied agents. Our experimental

results benchmarking both planning- and learning-based baselines illustrate the systematic evaluation that such a benchmark can provide for future efforts. We further perform an in-depth analysis of failure cases and provide insights for the future development of embodied social intelligence.

**Limitations**  While we aimed to preserve as much realism as possible, there are undoubtedly aspects of human behavior, particularly in how physical constraints manifest in the world, that are challenging to simulate. Meanwhile, the rule-based control of constrained agents makes their behavior lack diversity. This may be solved by leveraging LLMs to control constrained agents. Moreover, while we believe our challenge takes a good first step forward in introducing accessibility challenges to embodied social intelligence benchmarking efforts, we emphasize that our challenge is not representative of *all* possible constraints that such users may face.

## Acknowledgement

We thank Qinhong Zhou for his insightful feedback and help with paper writing, and Jeremy Schwartz and Esther Alter for setting up and updating the ThreeDWorld environments. This project is supported by the Honda Research Institute.

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

# A More Information about the CHAIC Challenge

## A.1 Benchmark Usage

The CHAIC Challenge can be accessed on the CHAIC GitHub as well as on its official webpage.

## A.2 Comparison with Other Embodied Challenges

We compare the differences between our proposed challenge and others in Table 3. Our CHAIC Challenge represents the first large-scale embodied social intelligence challenge focused on accessibility, incorporating outdoor scenes with emergent events, and requires goal inference from observations.

Table 3: **Comparison between Various Embodied Challenges.** *Social rearrangement assumes oracle position information of both agents and the target objects. **Smart Help assumes perfect perceiving of the other agents' actions and statuses.

| Challenge Name | Accessibility Setting | Multi-Agent Support | Goal Inference | Observation Type | Outdoor Scenes | Emergent Event |
|---|---|---|---|---|---|---|
| Watch-and-Help (Puig et al., 2021) | × | ✓ | × | Symbolic | × | × |
| NOPA (Puig et al., 2023a) | × | ✓ | ✓ | Symbolic | × | × |
| Social Rearrangement (Szot et al., 2023) | × | ✓ | × | Visual* | × | × |
| TDW-MAT (Zhang et al., 2023) | × | ✓ | × | Visual | × | × |
| Hazard (Zhou et al., 2024) | × | × | × | Visual | ✓ | × |
| Smart Help (Cao et al., 2024) | ✓ | ✓ | ✓ | Symbolic** | × | × |
| CHAIC (Ours) | ✓ | ✓ | ✓ | Visual | ✓ | ✓ |

# B Baseline Details

## B.1 More Details on LLM+BM Helper

We have tested several baselines in the benchmark, and the LLM+BM helper has shown the best performance. The LLM+BM helper consists of multiple modules. In addition to the behavior modeling module discussed in the main paper, the helper has several other modules, which we introduce here:

**Perception Module**  The perception module is used to extract useful information from raw RGB-D images. Following (Zhang et al., 2023), we have fine-tuned a Mask R-CNN (He et al., 2017) model on the images collected in the training dataset to obtain object-wise segmentation masks. The training set contains the same kinds of objects but with different layouts and scene backgrounds compared to the test set. The training details are in Appendix D.1.

**Memory Module**  The memory module is designed to understand the environment's layout and the positions of objects by an occupancy map and a semantic map. Firstly, the agent continuously updates a 2D grid-based top-down occupancy map when executing actions. Initially, all areas on the occupancy map are marked as unknown, and the map is updated using depth images. Utilizing depth images and camera intrinsics, the memory module first maps each pixel from the depth image into 3D space and then projects it onto the occupancy map. The semantic map, which also adopts a top-down view and maintains the same grid size as the occupancy map, records the locations of all detected objects within the grid.

**Decision Module**  An LLM-based decision module is used to generate a subgoal without any prior knowledge or specific design. The prompt encompasses six components: *task description*, *self-information*, *information about other agents*, *task progress*, *semantic map information*, and *available plans*. The LLM's output should be a plan from the list of valid plans, with specific object IDs included if the plan involves actions like "pick up" or "put on". Detailed information about the prompt is available in Appendix C.1.

**Execution Module**  The execution module is a low-level executor, which serves as a low-level executor, bridging the gap between high-level plans and low-level actions, including navigation and exploration. When the high-level plan directs the picking up of a previously seen object or traveling

to a goal space, the navigation module can recursively generate a path with low-level navigation commands using the occupancy map generated from the memory module. This map distinguishes between free, unknown, occupied, and wall spaces, assigning increasingly higher costs respectively. The navigation module employs the A* algorithm (Hart et al., 1968) to determine the most efficient route from the agent's current location to the target object and generates the necessary actions to follow this path. The path is recalculated whenever the occupancy map is updated. Additionally, for exploration tasks, the navigation module utilizes the frontier exploration method to enhance the efficiency of discovering new areas by repeatedly moving toward unknown spaces adjacent to known ones.

### B.2 Detailed Rules for Rule-Based Hierarchical Plan Helper (RHP)

The rule for Rule-Based Hierarchical Plan Helper (RHP) is similar to that of the constrained agent described in Section 5.1.1. The only difference is that since the helper does not know the exact goal of the constrained agent, the ruled-based agent will choose the target object randomly among all available objects.

In detail, at the start of an episode, the rule-based agent explores the environment to locate objects, containers, and the goal location. It picks up objects or containers if its hands are free. If over $50\%$ of the time steps remain and the agent does not obtain a container, it prioritizes acquiring one; otherwise, it focuses on objects. When unable to carry more, it deposits objects at the goal location. If less than $25\%$ of the time steps is left (or $37.5\%$ without a goal location identified), it immediately places objects in hands at the goal. The rule-based agent chooses the nearest object when multiple are available and puts objects in containers whenever possible.

## C  Prompt Details

### C.1  Detailed Prompt of Decision Module of LLM+BM Helper

The LLM+BM Helper uses an LLM-based decision module to determine the plan for the next step. The prompt of the LLM-based decision module contains six parts: *task description*, *self-information*, *information about other agents*, *task progress*, *semantic map information*, and *available plans*. The decision module needs to select a plan and fill in the plan with proper parameters, and then the execution module will execute the plan. Following are prompt descriptions and examples of each part:

#### C.1.1  Task Description

The **Task Description** includes a detailed description of the task's basic rules but does not explicitly show the constraints of the constrained agent. A prompt example is listed in Figure 4.

#### C.1.2  Self-Information

The **Self Information** contains information about the helper agent himself, including the previous actions of the helper with the statuses of these actions, the current position of the helper, and the objects that the helper is currently holding. A prompt example is listed in Figure 5.

#### C.1.3  Information about Other Agents

**Information about Other Agents** contains information about other agents, including the actions of the constrained agent that the helper has seen, together with their statuses, the objects that the helper has seen the constrained agent holding, the position of the constrained agent when the helper last saw him/her. For shopping tasks, it also includes the position of the child when the helper last saw her. A prompt example is listed in Figure 6.

#### C.1.4  Task Progress

The **Task Progress** contains all the objects that have been transported to the goal location, and the number of frames passed. A prompt example is listed in Figure 7.

**Task Description Prompt Example**

```
You are Bob. A constrained human David is walking a bike with his left hand
while accompanying his child. Your goal is to infer the target objects David
wants from his actions, and help him transport as many wanted target objects as
possible to <b03_fire_hydrant> (8882855) with the help of containers.

Note that:
- There are six shops in the environment, and the target objects are
distributed in these shops.
- David is accompanied by a bike and the bike has a basket, and David can put
at most three things into the bike basket, but David has to move the bike with
two hands and has to stop at a shop to put things into the basket. David moves
slow.
- You can hold two things at a time, and they can be either objects or
containers. You can put objects into the container (only after the container is
grasped) to hold more objects.
- All objects are identified by a unique name and ID, e.g. <table> (712).
- Actions cost several steps, and the maximum number of steps you can take is
3000. It may be costly to walk for long distance, so you need to transport
objects to the goal location as early as possible.
- A container can contain at most three objects, and will be lost once it is
transported to the goal location.
- Help David to supervise the child by following the child if she runs away.
- David is trying to get the same kind of things, so you should pick the things
that are of the same kind of the things that David picked.
```

Figure 4: Task Description Prompt Example

**Self Information Prompt Example**

```
Your previous actions and status are: [('moving at frame 0',
'ActionStatus.success'), ('pick up wood_basket <7157967> at frame 62',
'ActionStatus.success'), ('pick up grape <13682679> at frame 134',
'ActionStatus.success'), ('put the object in the container at frame 198',
'ActionStatus.success'), ('pick up apple <4455088> at frame 290',
'ActionStatus.success'), ('put the object in the container at frame 357',
'ActionStatus.success')]. Your current position is: (9.78, 3.26). You're
holding a container <wood_basket> (7157967) with target objects <grape>
(13682679), <apple> (4455088) in it.
```

Figure 5: Self Information Prompt Example

**Information about Other Agents Prompt Example**

```
David's previous actions and status are: [('pick up orange <8822607> at frame
433', 'ActionStatus.success')]. You have seen David holding these objects: a
bike with nothing in it, a bike with target object <apple> (15360225) in it, a
bike with target objects <apple> (15360225), <orange> (11935439) in it, a bike
with target objects <apple> (15360225), <orange> (11935439), <orange> (8822607)
in it. The last time you saw, David was at (3.63, 2.39) (meters). The last time
you saw, David's child was at (6.23, -0.2) (meters). David's target objects and
constraints should be infered from his actions and status.
```

Figure 6: Other Agents' Information Prompt Example

### C.1.5 Semantic Map Information

The **Semantic map information** contains all objects, containers, and the goal location information in the semantic map, with their position and height. A prompt example is listed in Figure 8.

```
Your current progress is: You've taken 1460/3000 steps. You have found the goal
position <b03_fire_hydrant> (14887175). You and David have already transported
<banana> (8826121), <banana> (11770901) to the <b03_fire_hydrant> (14887175).
```

Figure 7: Task Progress Prompt Example

```
You've seen these objects:  <orange> (11878369) is located at (8.86, -1.94)
(meters) with a height of 0.46 meters,  <apple> (15231642) is located at (9.26,
-1.98) (meters) with a height of 0.42 meters,  <apple> (9952058) is located at
(9.71, -2.0) (meters) with a height of 0.72 meters,  <orange> (10130190) is
located at (10.25, -1.97) (meters) with a height of 0.44 meters,  <orange>
(8108449) is located at (10.53, -1.91) (meters) with a height of 1.08 meters,
<apple> (14366820) is located at (10.7, 3.51) (meters) with a height of 0.77
meters,  <grape> (15240406) is located at (10.92, 3.48) (meters) with a height
of 1.03 meters,  <grape> (604227) is located at (10.88, 3.67) (meters) with a
height of 1.08 meters,  <croissant> (4072) is located at (13.88, -1.97)
(meters) with a height of 0.37 meters,  <burger> (9082737) is located at
(14.25, -1.98) (meters) with a height of 0.39 meters, container <wood_basket>
(5304525) is located at (14.21, -1.8) (meters) with a height of 1.13 meters,
container <plastic_basket> (10421069) is located at (14.09, 3.4) (meters) with
a height of 1.13 meters,  <b03_fire_hydrant> (14887175) is located at (1.73,
5.55) (meters) with a height of 0.4 meters.
```

Figure 8: Semantic Map Information Prompt Example

### C.1.6 Available Plans

The **Available Plans** contains all the available plans that the helper agent can take. A prompt example is listed in Figure 9.

```
Given your goal, previous actions, progress, and objects you see, please choose
the best action from the following action list to achieve your goal as soon as
possible: ['explore', 'follow child', 'turn around', 'transport object in hand
to goal space', 'goto and pick up target <orange> (11878369)', 'goto and pick
up target <apple> (15231642)', 'goto and pick up target <apple> (9952058)',
'goto and pick up target <orange> (10130190)', 'goto and pick up target
<orange> (8108449)', 'goto and pick up target <apple> (14366820)', 'goto and
pick up target <grape> (15240406)', 'goto and pick up target <grape> (604227)',
'goto and pick up target <croissant> (4072)', 'goto and pick up target <burger>
(9082737)', 'follow David']. Please choose one option from the list.
```

Figure 9: Available Plan Prompt Example

### C.2 Detailed Prompt of Decision Module of VLM Helper

The prompt of the decision module of VLM Helper is similar to that of LLM+BM Helper. The only difference is the VLM-based decision module perceives the behavior of the partner through raw RGB images as an additional observation. In detail, an image sequence with the last ten images is added to the input of the VLM-based decision module.

# D Perception Model Details

## D.1 Detection Model for Object Detection

Since the helper receives raw RGB-D images from the environment, an object detection model is necessary to identify objects within these images. We fine-tuned an object detection model using our dataset collected from training scenes.

**Data Collection**  To collect training data in the environment, a helper roams randomly and solely within the scenes, collecting egocentric images combined with ground truth segmentation. The environment is split into training and validation, and we collected 61K images at a resolution of $512 \times 512$ in total. There are 53 types of objects related to the benchmark, so the detection model has the same number of labels.

**Training Details**  We utilized the open-source code provided by MMDetection (Chen et al., 2019) as our training framework and selected a Mask R-CNN (He et al., 2017) model pre-trained on the COCO dataset (Lin et al., 2014) with a ResNet50 (He et al., 2016) backbone. The model was fine-tuned for four epochs, incorporating a warm-up stage of 500 steps and a batch size of 16. The optimizer employed was SGD with `lr = 0.01`, `momentum = 0.9`, and `weight_decay = 0.0001`. This fine-tuning process was finished on an NVIDIA A10G GPU in approximately six hours. The fine-tuned model achieved a 94.4% mAP@50 (Segmentation Mean Average Precision at 50% intersection over union) on the validation set.

## D.2 Action Recognition Model for Behavior Modeling

Recognizing the actions of the partner is a crucial ability for understanding its intentions, while current foundation models cannot directly discern actions in the wild. Therefore, an auxiliary action recognition model is necessary for our baseline agents. Similarly to the detection model, we fine-tuned an action recognition model.

**Data Collection**  Collecting behavior data is more challenging than gathering object detection data. To simulate the real situation, we created a follower whose sole action is to track the constrained agent. This follower has access to the action history, which when indicating that the constrained agent is acting, triggers the extraction of an RGB image sequence from the observer's viewpoint. These images are then concatenated into a video clip, each containing 50 to 100 frames at a resolution of $512 \times 512$. Sometimes the constrained agent is obscured, preventing full visibility throughout some actions. Consequently, we discarded any action clip where the visibility of the constrained agent was less than 20%. The dataset comprises six behaviors for the constrained agent: *successful pick-up, fail pick-up, put-in, successful put-on, fail put-on* and *moving*. In total, the dataset contains 3,000 video clips.

**Training Details**  We utilized the open-source tools provided by MMDetection (Contributors, 2020) for training and employed the Temporal Segment Network (TSN) (Wang et al., 2016), pre-trained on the Kinetics-400 dataset (Kay et al., 2017), as our base model with a ResNet50 backbone (He et al., 2016). The sampling strategy was set to $16 \times 1 \times 1$ (number of clips, clip length, clip interval). We fine-tuned the model 100 epochs using the same optimizer as object detection and selected the best checkpoint from the validation set. The fine-tuned model achieved 86.1% top-1 accuracy on the validation set.

# E More Results

## E.1 Additional Metrics

Besides the *transport rate (TR), Efficiency Improvement (EI)*, and *Goal Inference Accuracy (IA)*, we also calculated some meaningful metrics to measure the performance of helpers:

- **Completion Ratio of Helper (CR):** The proportion of tasks completed by the helper relative to the total number of completed tasks;

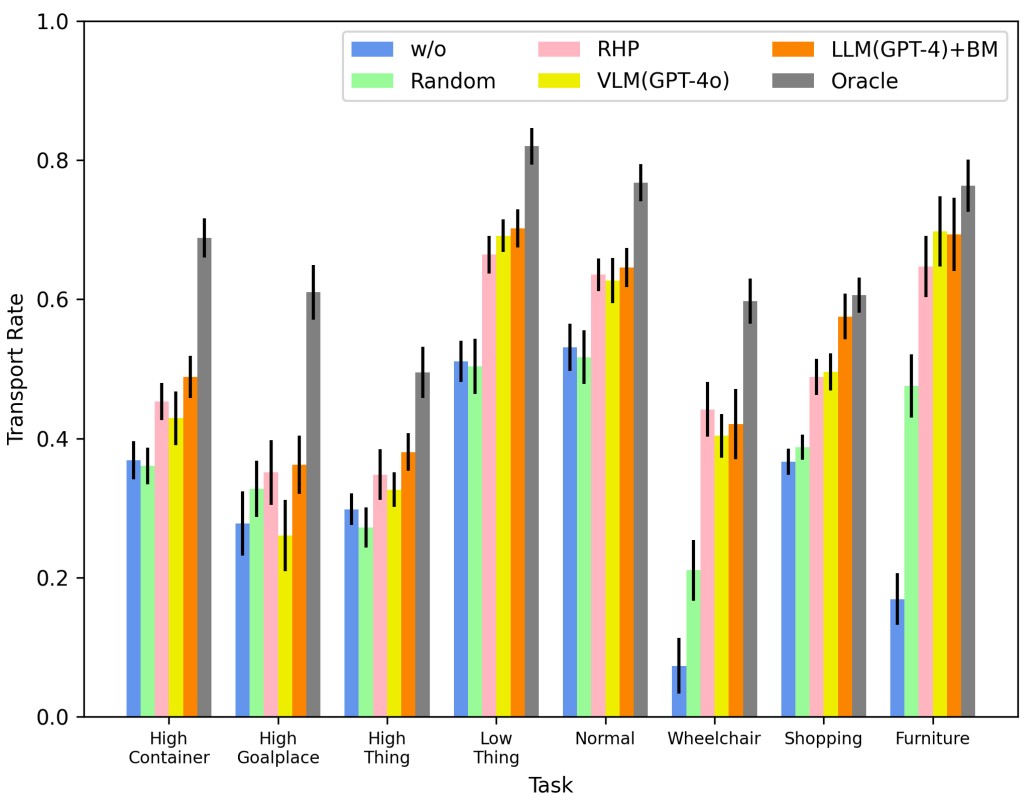

Figure 10: **Main Results with Error Bars:** The visualization of the transport rate with 1-sigma error bar of the standard error.

Table 4: **Additional results on CHAIC benchmark.** Here we report two more useful metrics: *Completion Ratio of Helper (CR)* and *Standard Error of Transport Rate (STD$_{TR}$)*. Commonly, a higher *CR* means the helper could do more parts in the tasks.

| | Indoor | | | | | | | |
|---|---|---|---|---|---|---|---|---|
| Helper Agent | Normal | | High Target | | High Container | | High Goalplace | |
| | CR↑ | STD | CR↑ | STD | CR↑ | STD | CR↑ | STD |
| w/o | / | 0.03 | / | 0.02 | / | 0.03 | / | 0.05 |
| Random | 0.09 | 0.04 | 0.10 | 0.03 | 0.12 | 0.03 | 0.06 | 0.04 |
| RHP | 0.15 | 0.02 | **0.43** | 0.04 | 0.29 | 0.03 | **0.39** | 0.05 |
| VLM (GPT-4o) | 0.13 | 0.03 | 0.08 | 0.02 | **0.34** | 0.04 | 0.18 | 0.05 |
| LLM (GPT-4) + BM | **0.22** | 0.03 | 0.30 | 0.03 | 0.30 | 0.03 | 0.35 | 0.04 |
| Oracle | 0.51 | 0.03 | 0.64 | 0.04 | 0.66 | 0.03 | 0.73 | 0.04 |

| | Indoor | | | | Outdoor | | | |
|---|---|---|---|---|---|---|---|---|
| Helper Agent | Low Target | | Obstacle | | Shopping | | Furniture | |
| | CR↑ | STD | CR↑ | STD | CR↑ | STD | CR↑ | STD |
| w/o | / | 0.03 | / | 0.04 | / | 0.02 | / | 0.04 |
| Random | 0.09 | 0.04 | 0.09 | 0.04 | 0.07 | 0.02 | 0.73 | 0.05 |
| RHP | 0.36 | 0.03 | 0.19 | 0.04 | 0.34 | 0.02 | 0.74 | 0.04 |
| VLM (GPT-4o) | **0.39** | 0.02 | 0.17 | 0.03 | 0.34 | 0.03 | **0.82** | 0.05 |
| LLM (GPT-4) + BM | 0.38 | 0.03 | **0.45** | 0.05 | **0.46** | 0.03 | 0.78 | 0.05 |
| Oracle | 0.59 | 0.03 | 0.38 | 0.03 | 0.45 | 0.03 | 0.77 | 0.04 |

Table 5: **Quantitative results of learning-based agents on CHAIC benchmark.** We report the average *Transport Rate (TR)* and *Efficiency Improvement (EI)* here. **w/o** means the main agent does the task solely without a helper.

| Indoor | | | | |
|---|---|---|---|---|
| Helper Agent | Normal
TR(EI)↑ | High Target
TR(EI)↑ | High Container
TR(EI)↑ | High Goalplace
TR(EI)↑ |
| w/o | 0.53 | 0.30 | 0.38 | 0.27 |
| RL | 0.45(-0.19) | 0.26(-0.16) | 0.28(-0.25) | 0.25(-0.22) |
| Smart-Help | 0.46(-0.12) | 0.24(-0.17) | 0.26(-0.28) | 0.31(0.01) |

| Indoor | | Outdoor | |
|---|---|---|---|
| Helper Agent | Low Target
TR(EI)↑ | Obstacle
TR(EI)↑ | Shopping
TR(EI)↑ | Furniture
TR(EI)↑ |
| w/o | 0.51 | 0.08 | 0.37 | 0.17 |
| RL | 0.43(-0.16) | 0.11(0.07) | 0.32(-0.13) | 0.67(0.74) |
| Smart-Help | 0.49(-0.04) | 0.13(0.11) | 0.32(-0.13) | 0.57(0.70) |

- **Standard Error of Transport Rate (STD$_{TR}$):** The standard error of transport rate among the test sets.

The results of these metrics are shown in Table 4.

## E.2 Error Bar of Transport Rate

We also visualize the 1-sigma standard error of transport rate for baseline helpers described in Section 5.2, shown in Figure 10.

## E.3 Comparison with Learning-Based Baselines

Although the baselines in the main paper are all non-training baselines, we also tested some learning-based baselines. The results are shown in Table 5. However, we observe that in most tasks, our learning-based baselines perform similarly to the main agent operating without a helper (except for the outdoor furniture task where two agents make a crucial difference because of their different strength capacities and the relatively easy task setting). This is primarily due to the following reasons:

- The inherent difficulty of vision-based reinforcement learning. Even with depth and segmentation information which we encoded as a semantic map, the agent is still hard to learn non-trivial features from the observations.

- The slow data collection process and the inability to parallelize in ThreeDWorld make it time-consuming to gather large-scale online data for training. In our experiments, collecting rollouts of size $10^4$ for each task requires one day on an NVIDIA A10G GPU.

**RL Baseline** An end-to-end reinforcement learning helper trained on each task separately. We use the Stable-Baselines3 (Raffin et al., 2021) codebase and PPO (Schulman et al., 2017) algorithm to wrap and train our RL helpers. We made minor modifications to the observation space to fit our training needs. Namely, we pack the RGBD image, semantic map, agent position/direction, and status of agent-holding objects as a customized RL observation. The reward is designed as a linear combination of transported objects and distance to the nearest target object. A penalty is also applied for each invalid action.

The policy network extracts features from each observation class with either CNN for images or MLP for scalar information and concatenates them. The concatenated features then pass through a two-layer MLP to produce a vector of length 64. This part is shared by both the actor and the critic in PPO. During training, we use a batch size of 2 and update the policy every 2 rollout step due to slow data collection. We use default training parameters in Stable-Baselines3, including $\gamma = 0.99, \lambda = 0.95, lr = 3 \times 10^{-4}$, etc. For each task, we train the model for $10^4$ steps.

**SmartHelp Baseline** A helping method proposed by Cao et al. (2024). For the opponent modeling module, we set the window size $w$ of the state feature to 5. The input of the opponent modeling is the observation of the helper agent, which contains information on each object in the helper's view, as well as the constrained agent's information, if visible to the helper. The information on each object includes the type, weight, position, and height. The information on the constrained agent includes position and orientation, the last action with its status, and the objects currently held by the agent. The helper receives the ground truth actions and statuses of the constrained agent.

We run simulations with a random helper and an oracle constrained agent for 12 episodes each to collect the dataset of constrained agent trajectories used for training the opponent model. We collect 8355 trajectories after balancing, each containing observations at five discrete time points, together with the constrained agent's goal and constraint. The constraint of the constrained agent is a 5-dimension vector, which includes the maximum and minimum heights the agent can reach, the maximum weight the agent can hold, whether the agent is holding a bike and whether the agent is confined to a wheelchair, with each dimension scaled to [0, 1]. The goal contains the type of the goal and possible target index. There are six types of goal: "explore", "wait", "pick", "puton", "putin", "unknown". If the helper agent doesn't see the constrained agent for all five observations, the ground truth goal is set to "unknown". We balance the number of data for each goal in the trajectory dataset. For some goals like "pick", there is a target index for the goal indicating the type of the object it picks, and there are 53 types of objects in total. We trained the opponent model using the cross-entropy loss for goals and the mean squared error loss for constraints. We use the Adam optimizer with a learning rate of $1 \times 10^{-6}$ and a batch size of 32. Finally, we achieved the goal prediction accuracy of 93%, and the mean squared error loss for the constraint is 0.02.

After training the opponent modeling module, we use the same setting of the **RL Baseline** to train a policy module for $10^4$ steps on each task.

# F Additional Qualitative Analysis of LLM+BM and VLM Helper Behaviors

## F.1 LLM+BM Helper Behaviors

We analyze the chain-of-thought outputs of LLM and the actions of LLM+BM Helper, and Figure 11 is an example of the shopping task.

**LLM+BM helper knows to use containers to increase transport efficiency** As shown in Figure 11a, at the beginning of the task, the helper did not see David, the constrained agent, pick anything yet. But he could reason to pick up the container to improve transportation efficiency. He saw two containers when he was planning and chose the container closer to his current position.

**LLM+BM helper can infer objects that constrained agent needs correctly** As shown in Figure 11b, after seeing David picking up an apple, the helper could reason out that David wanted fruits, so he picked up an apple, and he could also pick up other fruits like grapes afterward.

**LLM+BM helper knows to put things in container continuously** As shown in Figure 11c, after picking up the apple, the helper reasoned to put the apple into the container, freeing up one hand to pick up more target objects.

**LLM+BM helper knows when to transport the objects into goal space** As shown in Figure 11d, after putting three objects into the container and picking an object in the other hand, the helper decided to transport them to goal space, the fire hydrant in front of the house of David's. The helper analyzed that he could not get objects anymore while transporting these objects with maximum efficiency, and David needed the objects he transported.

## F.2 VLM Helper Behaviors

We also analyzed some behaviors of the VLM Helper, and Figure 12 is an example of the high container task.

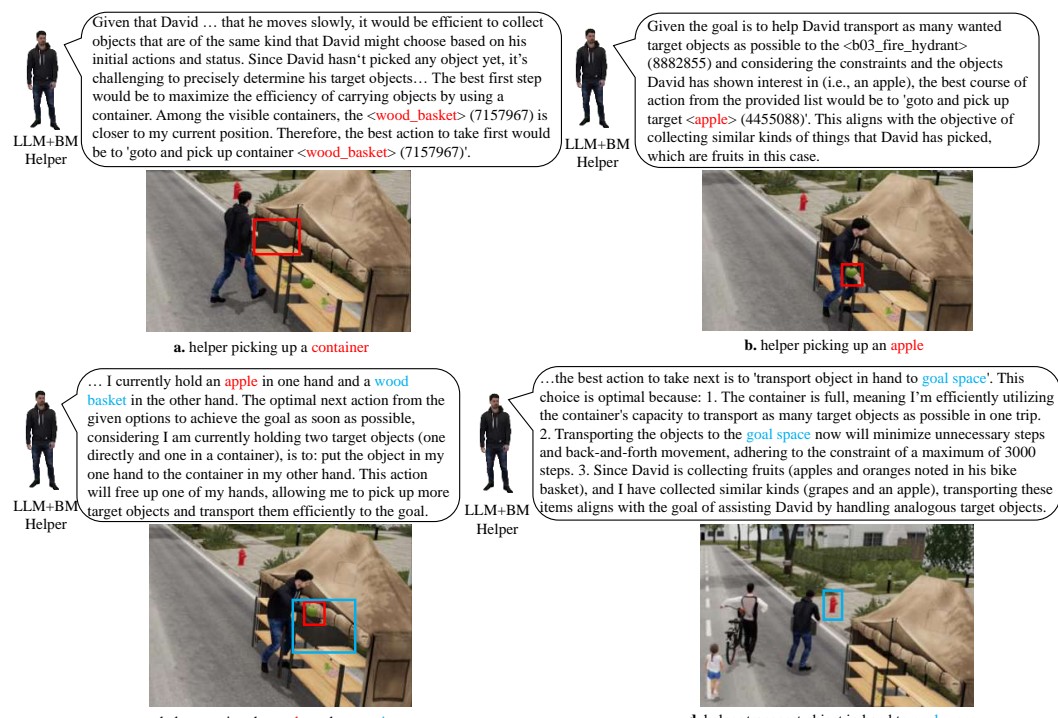

a. helper picking up a container
b. helper picking up an apple
c. helper putting the apple to the container
d. helper transport object in hand to goal space

Figure 11: LLM+BM Helper's behaviors in one episode of the shopping task, together with the chain-of-thought outputs of the LLM-based decision module (Some of the outputs are omitted for space reasons).

**VLM Helper knows to explore at first**    As shown in Figure 12a, at the beginning of the task, the helper could not see the constrained agent. Therefore, the helper chose to explore to find the constrained agent and objects.

**VLM Helper knows to follow the constrained agent**    As shown in Figure 12b, after seeing the constrained agent, the helper chose to follow her to get the information about the things she needed. Then, he saw the constrained agent pick up bread. However, sometimes the VLM helper is unable to infer the objects needed by the constrained agent, causing him to consistently follow the constrained agent.

**VLM Helper sometimes infer objects that constrained agent needs correctly**    As shown in Figure 12c and Figure 12d, after seeing the constrained agent picking bread, the helper could collect other objects of this kind, like loaf bread and hamburger.

**VLM Helper cannot transport objects efficiently**    In this episode, the VLM Helper only transported one thing at a time to the goal space, which is inefficient. First, the helper didn't use the container. Second, the helper didn't use both hands to carry the target objects. The most efficient way is to hold one container with three objects in one hand and hold one object in the other hand, thus transporting four objects at a time.

# G   Details of Outdoor Task Generation

## G.1   Details of Shopping Task Generation

For the shopping task, six shops are generated and spread out on both sides of the road. Each shop sells one specific category of items. There are three categories of items, each of which is sold in exactly two stores. The item categories and specific items for each category are listed below:

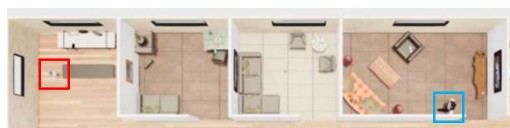 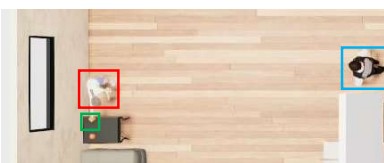

**a.** helper explores at first      **b.** helper seeing constrained agent picking bread

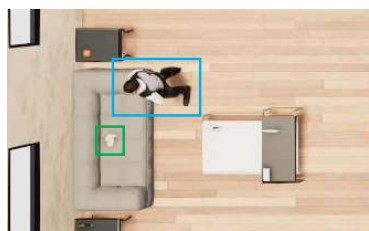 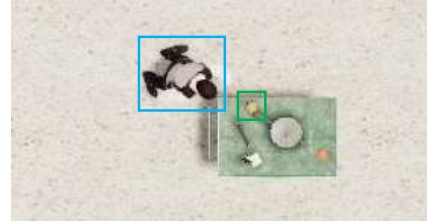

**c.** helper picking loaf bread      **d.** helper picking hamburger

Figure 12: VLM Helper's behaviors in one episode of the high container task

- Fruit: apple, orange, grape, and banana.
- Baked food: loaf bread, croissant, burger, and donut.
- Drink: cola, pepsi, sprite, and fanta.

For each episode, we first randomly select one category of items. Then randomly select several objects in this category, together with another object randomly selected in the other two categories as the target objects. The number of target objects is between 11 and 13. The goal location is a fixed, pre-determined place in front of the bicycle agent's house. Then we randomly select three shops and put a container on each of them. Finally, we randomly initialize the two agents. Their initial positions are guaranteed at least 0.5 meters away from the nearest shop.

### G.2 Details of Moving House Task Generation

We choose twelve common pieces of furniture for the moving house task. For each episode, we randomly select five pieces of furniture and randomly put them in the area in front of the house. We ensure that the furniture does not overlap each other. Then we set the initial positions of the agents near the place area. The goal location is a truck parked around 10 meters away from the place area. The frail agent's strength is 100, while the helper agent's strength is 600. The weights of furniture range from 50 to 900.

## H  Statement

### H.1  Broader Impacts

By building the CHAIC benchmark, our work tries to simulate the disability of human beings in a simulated environment. The benchmark proposes several tasks and scenes that are common in the real world. After setting up the benchmark, we built a few helper agents to help the constrained agents fulfill their tasks with visual observation only. This kind of helper agent has a wide range of potential usage in real life but there is not much research on this in academia. Through this research, we hope to pave the way to more friendly and helpful AI agents that can be implemented in both simulated environments and the real world.

**Potential Negative Impacts**  While some baselines are driven by foundation models and achieving the best scores in our experiments, applying them may generate malicious content and potentially

lead to bad actions. It is important to set up mechanisms to detect such actions before these helpers can be put into real-world usage.

## H.2   Responsibility

The authors declare that they bear full responsibility for any violations of rights associated with this dataset.

## H.3   LICENSE

The CHAIC benchmark is licensed under the MIT license. Meanwhile, the benchmark is built upon several open-source projects, and we list their licenses here:

- ThreeDWorld: BSD-2-Clause license
- MMAction2: Apache-2.0 license
- MMDetection: Apache-2.0 license

## H.4   Resubmission Discussion

The paper is withdrawn from CVPR2024, where the reviewers posted two main points:

- The reviewers thought it was not natural that the helper could know the whole action history of the constrained agent. We agree with it, and currently, the helper cannot get any text information about the action history of the constrained agent and needs to infer it from raw RGB-D observation.
- The reviewers thought the objects and tasks were not rich enough. Therefore, we create both indoor and outdoor scenes with various tasks in this submission, and the number of task-relevant objects increases from about 20 to over 50.

## I   Datasheets

### I.1   Motivation

- **For what purpose was the dataset created? Was there a specific task in mind? Was there a specific gap that needed to be filled? Please provide a description.**
  The dataset was established to research human-AI cooperation in embodied and realistic settings.
- **Who funded the creation of the dataset?** If there is an associated grant, please provide the name of the grantor and the grant name and number.
  N/A.
- **Any other comments?**
  No.

### I.2   Composition

- **What do the instances that comprise the dataset represent (e.g., documents, photos, people, countries)?** Are there multiple types of instances (e.g., movies, users, and ratings; people and interactions between them; nodes and edges)? Please provide a description.
  One instance is a sequence of commands and the ThreeDWorld Platform can read the commands in the instance and then create the scenes and tasks.
- **How many instances are there in total (of each type, if appropriate)?**
  There are 192 instances in total. There are 8 tasks, and each task contains 12 training instances and 12 testing instances.
- **Does the dataset contain all possible instances or is it a sample (not necessarily random) of instances from a larger set?** If the dataset is a sample, then what is the larger set? Is the sample representative of the larger set (e.g., geographic coverage)? If so, please describe

how this representativeness was validated/verified. If it is not representative of the larger set, please describe why not (e.g., to cover a more diverse range of instances, because instances were withheld or unavailable).

The dataset is a sample. We have an instance generation pipeline that can generate infinite instances for each task.

- **What data does each instance consist of?** "Raw" data (e.g., unprocessed text or images) or features? In either case, please provide a description.

  Each instance consists of a command sequence used for task initialization in the ThreeDWorld Platform.

- **Is there a label or target associated with each instance?** If so, please provide a description.

  No.

- **Is any information missing from individual instances?** If so, please provide a description, explaining why this information is missing (e.g., because it was unavailable). This does not include intentionally removed information, but might include, e.g., redacted text.

  No.

- **Are relationships between individual instances made explicit** If so, please describe how these relationships are made explicit.

  No.

- **Are there recommended data splits (e.g., training, development/validation, testing)?** If so, please provide a description of these splits, explaining the rationale behind them.

  The training-testing split in the dataset has a ratio of 1:1.

- **Are there any errors, sources of noise, or redundancies in the dataset?** If so, please provide a description.

  No.

- **Is the dataset self-contained, or does it link to or otherwise rely on external resources (e.g., websites, tweets, other datasets)?** If it links to or relies on external resources, a) are there guarantees that they will exist, and remain constant, over time; b) are there official archival versions of the complete dataset (i.e., including the external resources as they existed at the time the dataset was created); c) are there any restrictions (e.g., licenses, fees) associated with any of the external resources that might apply to a future user? Please provide descriptions of all external resources and any restrictions associated with them, as well as links or other access points, as appropriate.

  The dataset is linked to the ThreeDWorld Platform, a) which has been maintained and will be maintained for a long time at TDW GitHub, and we use its stable version 1.12.27. b) the complete dataset can be found at CHAIC GitHub. c) The 3D models used in the dataset can be publicly downloaded via Google Drive. The dataset is subjected to an MIT License.

- **Does the dataset contain data that might be considered confidential (e.g., data that is protected by legal privilege or by doctor-patient confidentiality, data that includes the content of individuals' non-public communications)?** If so, please provide a description.

  No.

- **Does the dataset contain data that, if viewed directly, might be offensive, insulting, threatening, or might otherwise cause anxiety?** If so, please describe why.

  No.

- **Does the dataset relate to people?** If not, you may skip the remaining questions in this section.

  No. Only humanoid agents are used in the dataset.

- **Any other comments?**

  No

## I.3 Collection Process

- **How was the data associated with each instance acquired?** Was the data directly observable (e.g., raw text, movie ratings), reported by subjects (e.g., survey responses), or

indirectly inferred/derived from other data (e.g., part-of-speech tags, model-based guesses for age or language)? If data was reported by subjects or indirectly inferred/derived from other data, was the data validated/verified? If so, please describe how.

The data is not directly observable or reported. Data is created from an automatic scene generation pipeline and visualized via the ThreeDWorld Platform.

- **What mechanisms or procedures were used to collect the data (e.g., hardware apparatus or sensor, manual human curation, software program, software API)?** How were these mechanisms or procedures validated?

  We use an automatic scene generation pipeline for each task to generate data.

- **If the dataset is a sample from a larger set, what was the sampling strategy (e.g., deterministic, probabilistic with specific sampling probabilities)?**

  Random Sampling.

- **Who was involved in the data collection process (e.g., students, crowdworkers, contractors) and how were they compensated (e.g., how much were crowdworkers paid)?**

  Since we automatically generated data, only authors were involved in the data collection process.

- **Over what timeframe was the data collected?** Does this timeframe match the creation timeframe of the data associated with the instances (e.g., recent crawl of old news articles)? If not, please describe the timeframe in which the data associated with the instances was created.

  The data was collected between 15 May and 10 June 2024.

- **Were any ethical review processes conducted (e.g., by an institutional review board)?** If so, please provide a description of these review processes, including the outcomes, as well as a link or other access point to any supporting documentation.

  No.

- **Does the dataset relate to people?** If not, you may skip the remainder of the questions in this section.

  No.

- **Any other comments?**

  No.

## I.4 Preprocess/cleaning/labeling

- **Was any preprocessing/cleaning/labeling of the data done (e.g., discretization or bucketing, tokenization, part-of-speech tagging, SIFT feature extraction, removal of instances, processing of missing values)?** If so, please provide a description. If not, you may skip the remainder of the questions in this section.

  No.

- **Any other comments?**

  No

## I.5 Uses

- **Has the dataset been used for any tasks already?** If so, please provide a description.

  No.

- **Is there a repository that links to any or all papers or systems that use the dataset?** If so, please provide a link or other access point.

  No.

- **What (other) tasks could the dataset be used for?**

  Embodied behavior recognition, embodied object detection.

- **Is there anything about the composition of the dataset or the way it was collected and preprocessed/cleaned/labeled that might impact future uses?** For example, is there anything that a future user might need to know to avoid uses that could result in unfair

treatment of individuals or groups (e.g., stereotyping, quality of service issues) or other undesirable harms (e.g., financial harms, legal risks) If so, please provide a description. Is there anything a future user could do to mitigate these undesirable harms?

No.

- **Are there tasks for which the dataset should not be used?** If so, please provide a description.

  No.

- **Any other comments?**

  No

## I.6 Distribution

- **Will the dataset be distributed to third parties outside of the entity (e.g., company, institution, organization) on behalf of which the dataset was created?** If so, please provide a description.

  Yes, the dataset is public and everyone is welcome to use it.

- **How will the dataset will be distributed (e.g., tarball on website, API, GitHub)? Does the dataset have a digital object identifier (DOI)?**

  The dataset will be distributed via GitHub. Currently, the dataset does not have a DOI.

- **When will the dataset be distributed?**

  Before NeurIPS 2024.

- **Will the dataset be distributed under a copyright or other intellectual property (IP) license, and/or under applicable terms of use (ToU)?** If so, please describe this license and/or ToU, and provide a link or other access point to, or otherwise reproduce, any relevant licensing terms or ToU, as well as any fees associated with these restrictions.

  This dataset is licensed under the MIT License.

- **Have any third parties imposed IP-based or other restrictions on the data associated with the instances?** If so, please describe these restrictions, and provide a link or other access point to, or otherwise reproduce, any relevant licensing terms, as well as any fees associated with these restrictions.

  No.

- **Do any export controls or other regulatory restrictions apply to the dataset or to individual instances?** If so, please describe these restrictions, and provide a link or other access point to, or otherwise reproduce, any supporting documentation.

  No.

- **Any other comments?**

  No.

## I.7 Maintainance

- **Who will be supporting/hosting/maintaining the dataset?**

  Our CHAIC Team will maintain the dataset.

- **How can the owner/curator/manager of the dataset be contacted (e.g., email address)?**

  We will release the non-anonymous version after review.

- **Is there an erratum?** If so, please provide a link or other access point.

  There is no erratum yet, and the future erratum will be released in the GitHub repo if any.

- **Will the dataset be updated (e.g., to correct labeling errors, add new instances, delete instances)?** If so, please describe how often, by whom, and how updates will be communicated to dataset consumers (e.g., mailing list, GitHub).

  This will be updated to GitHub when we find errors or make improvements.

- **If the dataset relates to people, are there applicable limits on the retention of the data associated with the instances (e.g., were the individuals in question told that their data**

**would be retained for a fixed period of time and then deleted)?** If so, please describe these limits and explain how they will be enforced.

N/A.

- **Will older versions of the dataset continue to be supported/hosted/maintained?** If so, please describe how. If not, please describe how its obsolescence will be communicated to dataset consumers.

  Yes, the old version will be kept in GitHub.

- **If others want to extend/augment/build on/contribute to the dataset, is there a mechanism for them to do so?** If so, please provide a description. Will these contributions be validated/verified? If so, please describe how. If not, why not? Is there a process for communicating/distributing these contributions to dataset consumers? If so, please provide a description.

  Others can contact paper authors when they want to contribute.

- **Any other comments?**

  No.

