# OpenReview forum: "Constrained Human-AI Cooperation: An Inclusive Embodied Social Intelligence Challenge"
_NeurIPS.cc/2024/Datasets_and_Benchmarks_Track — NeurIPS 2024 Track Datasets and Benchmarks Poster_

### Official Review · Reviewer_xLSK · 2024-06-25
**Good paper for further advancing HMT in accessibility scenarios**

**Rating:** 7
**Confidence:** 3
**Clarity:** The paper is well-written.

**Review:**

While limited in the number of constraints, the challenge introduced here can be extended to more scenarios and is a significant contribution to human-AI cooperation and planning in constrained tasks. It differs from previous solutions by incorporating a more realistic stimulus based on visual perception instead of symbolical input. It includes outdoor scenes and perturbations, moving the scope of evaluation closer to real-world scenarios.

The authors use their challenge to benchmark models, including a novel agent incorporating traditional recognition models for visual perception and LLMs for planning. While the results of LLM-empowered helpers are not much better than those of a simpler RHP, they constitute important baselines for future work and can guide the development of more socially aware agents.

**Strengths:**

- The authors propose a new challenge for evaluating socially-aware intelligent agents assisting humans with physical constraints.
- The authors benchmark different models on the task, including LLM-based ones with state-of-the-art models.
- The authors incorporate scenarios with indoor and outdoor scenes and perturbations.

**Additional Feedback:**

None.

**Correctness:**

Design and evaluation were performed correctly. The appendix contains additional results.

**Documentation:**

Except for the points reported in "Opportunities for Improvement", the work has enough details for reproducibility. The appendix is rich, the project website contains videos demonstrating the agents in the environment, and the Repo includes instructions on executing the code.

**Ethics:**

No ethical concerns.

**Limitations:**

The authors have adequately addressed the limitations and potential negative societal impact of their work.

**Opportunities For Improvement:**

Some details are missing, such as:

- It is unclear whether the helper agent moves the object to the target locations by itself or to an area where the primary agent can get the objects and move them to a target location by themselves. If the answer is the former, how does the helper agent know where to move the target object since the primary agent is the only one that knows about the target location?
- Given that the primary agent's and helper agent's actions are asynchronous, I assume the primary agent can move to pursue another object while the helper agent gets an out-of-reach object or removes obstacles for the principal agent. In that case, the leading agent can get out of the sight of the helper agent. Does the helper agent follow the leading agent afterward? Does it know the location of the principal agent at all times?
- Videos used for training end when an agent starts a new action (line 101 in D.2). What are the decision points during test time? Does the helper agent know that the primary agent is executing an action and just has to infer which one, or does it also have to infer whether the primary agent is executing an action from the live stream?

**Relation To Prior Work:**

The authors clearly discuss how this work differ from previous contributions in the main body of the paper and summarize the main differences in Table 1 in the appendix.

**Summary And Contributions:**

This paper introduces a social intelligence challenge that embodies physically constrained humans trying to perform some task. The environment comprises a leading agent, a helper agent, obstacles, objects, and locations. The goal is to make the helper agent assist the primary agent in transporting a series of target objects from random locations to a goal location. The helper agent must recognize the primary agent's goals and constraints from an egocentric visual input. In some conditions, perturbations can occur (e.g., a child running away from the principal agent), which require the helper to adapt its current strategy.

---

> ### Author Rebuttal · Authors · 2024-08-17
>
> We greatly appreciate your encouraging remarks and helpful suggestions on our work. Here is the explanation for the missing details you mentioned in the opportunity for improvement section.
>
> > Q1: It is unclear whether the helper agent moves the object to the target locations by itself or to an area where the primary agent can get the objects and move them to a target location by themselves.
> >
> - We apologize for any misunderstandings that may have occurred in our paper. In our setting, the name of the target location is shown to both the constrained agent and the helper agents, while the names of the objects that the constrained agent needs are hidden for the helper agents.
> - Both ways to move the objects are acceptable for the helper. It can **put objects on the ground or some table** and let the constrained agent transport them or **transport them directly to the target location**.
>
> > Q2: The leading agent can get out of the sight of the helper agent. Does the helper agent follow the leading agent afterward? Does it know the location of the principal agent at all times?
> >
> - In our setting, the helper can only get the location of the principal (constrained) agent when it is in the helper's view.
> - *‘Following another agent’* is an available plan for LM-based agents. When the helper is undergoing this plan, it first tries to find this agent using **frontier exploration** (i.e., repeatedly walking to a nearby known part). After seeing another agent, the helper will **keep watching** it at a certain distance (3m).
> - Sometimes, another agent gets out of the helper's sight. Then, the helper's low-level executor will perform a sequence of pre-defined low-level actions: **go to** the position where the principal agent last appeared and then **turn around** until finding another agent. The helper will return to the frontier exploration stage if another agent is still out of sight.
>
>     When it finishes one plan, the helper can choose whether to follow another agent. It can choose ‘*following another agent’* as its next plan or take other actions.
>
>
> > Q3: Videos used for training end when an agent starts a new action (line 101 in D.2). What are the decision points during test time? Does the helper agent know that the primary agent is executing an action and just has to infer which one, or does it also have to infer whether the primary agent is executing an action from the live stream?
> >
> - Here, the helper needs to infer whether the primary agent is **executing an action** and **figure out the action's name** from the live stream.
> - In our LLM+BM helper agent implementation, the helper calls the action recognition model every few steps. When the probability that the primary agent is performing a specific action exceeds a threshold, the LLM+BM helper recognizes that the primary agent is executing this action.
>
> We hope further clarification on the missing parts clarifies our contribution. If you have any more questions, please feel free to let us know during the rebuttal window. We appreciate your suggestions and comments!

---

> ### Comment · Reviewer_xLSK · 2024-08-22
> **Keeping my rating**
>
> Thank you for addressing my concerns. I still believe this is a good paper.

---

### Official Review · Reviewer_MFUo · 2024-07-24
**Review of CHAIC**

**Rating:** 7
**Confidence:** 3
**Correctness:** Yes
**Clarity:** Yes

**Review:**

Within the challenge situation, current VLM and LLM do not properly infer solutions to successfully transport target objects with a great gap with the upper bound performance. Customized behavior modeling module also shows enhanced performance compared to baselines. Overall documentation is well-organized.

**Strengths:**

- Handling both indoor/outdoor accessibility, which is an important physical constraint happening in the real world.
- Detailed documentation of the whole process in supplementary materials.
- Proper baselines from rule-based to learning-based methods.

**Additional Feedback:**

N/A

**Documentation:**

Yes

**Limitations:**

Yes

**Opportunities For Improvement:**

- Need more specification of metric in section 3.1: How is a successful rate designed? Are there any references for designing it? Moreover, how constants (i.e., alpha and beta) are decided?

**Relation To Prior Work:**

Yes

**Summary And Contributions:**

This paper proposes an embodied challenge in indoor/outdoor social situations where physically-constrained agents need to achieve goals with the help of other agents. Four types of constrained agents are defined, and helper agents should infer what to do in a given situation.

---

> ### Author Rebuttal · Authors · 2024-08-17
>
> Thank you for your insightful comments and positive feedback on our work.
>
> We would like to clarify your suggestion for more specifications on the metrics and constraints here and in our paper's revision.
>
> > Q1: Need more specification of metric in section 3.1: How is a successful rate designed? Are there any references for designing it? Moreover, how constants (i.e., alpha and beta) are decided?
> >
> - Each agent has two main properties: reaching range and strength. The **reaching range** is determined based on the agent's model height and status. For example, a little child struggles to reach high objects, and a person in a wheelchair has difficulty reaching the ground. The **strength** is based on its agent model. Intuitively, objects with light models can be picked up quickly, but those with heavy models are hard to pick up. If an action can be carried out without violating constraints, its success rate is $100$%.
> - Two parameters, $\alpha$ and $\beta$, act as soft constraints. These allow the agent to **occasionally perform actions beyond its typical capabilities**, similar to how a child agent might successfully reach an apple on top of a cupboard by jumping; sometimes, it can succeed, and sometimes, it cannot.  The success rate for these actions is modeled using **an exponential probability function**, $\exp(-\delta / \alpha) / \beta$. Here, $\delta$ is the excess part of the constraint, $\alpha$ normalizes the impact of each type of constraint (range and strength) to the same scale, and $\beta$ is set to $2$ for the reaching range constraint and $1$ for the strength constraint, reflecting the discontinuous nature of jumping compared to the continuous nature of carrying. If multiple constraints are broken, the overall success rate of this action will be the multiplication of each success rate.
>
> We hope that providing further details clarifies our contribution. Should you have any additional questions, please feel free to raise them during the rebuttal period. We value your suggestions and feedback! Thanks again for your positive review.

---

### Official Review · Reviewer_8256 · 2024-07-25
**Review of Submission867**

**Rating:** 6
**Confidence:** 3

**Review:**

As the authors point out, this is the first large-scale embodied social intelligence challenge with accessibility explicitly in mind --- which is likely an important aspect to focus on considering the graying of populations in countries across the world. However, I think the paper needs a few revisions before being suitable for publication.

**Post-rebuttal update**

I am raising my score from 5 to 6 after the rebuttal.

**Strengths:**

**Significance**: The task is interesting and well-motivated.

**Quality**
- The authors consider a variety of agents, including rule-based and SOTA LLM-based agents.
- The authors provide a detailed datasheet in the appendix.

**Social implications**: The work is likely to have a positive social impact.

**Additional Feedback:**

## Questions for the authors

- Lines 127--130: Why is the reaching range provided for the Child and Wheelchair agents, but not the Bicycle and Frail agents?
- Line 121: 'successful rate': Do you mean 'success rate'"?
- Lines 121--122: What do mean by 'the excess part'?
- Which of the actions in the action space are relevant to the emergency events? How does the helper agent stop the child from running away?
- Table 1: 'Emergent event' -> Should this be 'Emergency event' instead?
- Line 298: 'Object spaceship' -> What does this mean?

**Clarity:**

The paper could use a thorough proofreading/copyediting pass.

## Typos

- Line 107: Add a space before the '('
- Line 133: Should the colon be a period instead?
- Line 145: Should there be an 'and' after the comma?
- Line 210: 'handwriting' -> 'handwritten'
- Line 217: 'on goal place' -> 'on the goal place'
- Line 251: 'appendix' -> 'the appendix'
- Line 266: 'out-of-distribute' -> 'out-of-distribution'
- Line 291: 'Emergent Rate' -> 'Emergency Rate'
- Line 292: 'even LLM' -> 'even if LLM'

## Suggestions

- Line 137: The symbol $T$ is used for max timesteps later in the paper (lines 176--177). I recommend using a different symbol...
- Line 158: 'high target object' -> 'High target' (to be consistent with Table 1)
- Line 168: 'moving furniture' -> 'Moving house' (to be consistent with Table 1)
- I think it would be better to say 'constrained agent' (which you do have in Figure 3 and line 308) rather than 'constraint agent'.
- Table 2: 'Low Thing' -> 'Low Target' (to be consistent with Table 1)
- Use the `booktabs` package for better-looking tables: https://ctan.org/pkg/booktabs?lang=en

## Unclear writing

- Line 215: 'Since it is often a long walk to go to the goal place.' --> This sentence seems like a non-sequitur. It's not clear if this is supposed to connect to the sentence before it or after it.

**Correctness:**

The dataset seems constructed in a sound way. However, it is a bit difficult to trust the experimental results without error bars.

**Documentation:**

## Dataset

There is sufficient detail on data collection and organization, availability and maintenance, and ethical and responsible use.

## Benchmark

- The documentation provided in the repo's README.md describes how to reproduce the results in the paper, which is good. However, there is no guidance on how another researcher would go about creating and evaluating a new agent for this challenge.
- Line 238: I could not find details on the designed rules in the Appendix.

**Ethics:**

I do not have any ethical concerns with this paper.

**Limitations:**

The authors have adequately addressed the limitations and potential negative societal impact of their work.

**Opportunities For Improvement:**

- The lack of error bars for the results makes it harder to trust the claims that the authors are making, especially in some cases when the scores for the different agents are close to each other (e.g., the TR score for the Normal environment with the LLM+BM agent is 0.66 while the TR score for the VLM agent is 0.65). The authors marked 'Yes' for question 7 in the NeurIPS checklist, but I did not see any error bars in the appendix.

**Relation To Prior Work:**

Yes, the authors clearly discuss how this work differs from previous contributions.

**Summary And Contributions:**

This paper presents a new benchmark for evaluating AI agents' capability to help 'constrained agents', e.g., humans with disabilities, children, etc. The benchmark involves long-horizon tasks in a 3D environment. The authors evaluate both rule-based and learning-based agents on these tasks, including an agent with an explicit behavior modeling (BM) module, and find that the LLM+BM agent achieves the best performance on the majority of metrics and tasks.

---

> ### Author Rebuttal · Authors · 2024-08-17
>
> Thank you for your thorough evaluation and insightful review of our work. Your comments and suggestions have been very helpful in our revision process. We address each of your comments and questions one by one in the following paragraphs:
>
> > Q1: The lack of error bars for the results makes it harder to trust the claims that the authors are making.
> >
> - Although we do not draw the error bar explicitly, we report **the standard deviation** of each method in Appendix Table 2. Meanwhile, we conducted additional experiments for the eight tasks to **draw the 1-sigma error bar of the standard error** in the PDF file attached to the global rebuttal. The error bar represents the standard error over 24 seeds (2 seeds per scene, 12 scenes) for each task. We assume normally distributed errors of TR. As shown in the PDF, with additional runs, our **main findings in the paper still hold.**
> - Besides, if we take the average of TR among all eight tasks, there is a **notable performance gap between the LLM+BM and VLM agents** (52.75 vs 48). Therefore, at the comprehensive level, the LLM + BM agent is better than the VLM agent.
>
> > Q2: The paper could use a thorough proofreading/copyediting pass.
> >
> - Thank you for your careful reading and suggestions. We apologize for the typos and unclear writing in the paper and accept all your suggestions about clarity. We will proofread our revision thoroughly again.
> - Regarding the unclear writing you mentioned, the sentence *'Since it is often a long walk to go to the goal place.'* is connected to the next sentence, and we will reorganize the sentence to clarify it in the revision.
>
> > Q3: However, there is no guidance on how another researcher would go about creating and evaluating a new agent for this challenge.
> >
> - Thanks for this suggestion. We **added a section named ‘creating a new agent’** to the guide for creating and evaluating a new agent to README in GitHub.
>
> > Q4: Line 238: I could not find details on the designed rules in the Appendix.
> >
> - We apologize for this missing part in the Appendix. We will add the description to the Appendix.
> - To clarify in the interim, the rule for the rule-based agent is similar to that of the constrained agent described in Section 5.1.1. The only difference is that since the helper does not know the exact goal of the constrained agent, the ruled-based agent will choose the target object randomly among all available objects.
> - In detail, at the start of an episode, the rule-based agent explores the environment to locate objects, containers, and the goal place. It picks up objects or containers if its hands are free. If over $50$% of the time remains and lacks a container, it prioritizes acquiring one; otherwise, it focuses on objects. When unable to carry more, it deposits objects at the goal places. If less than $25$% of the time remains (or $37.5$% without a goal location identified), it immediately places objects in hands at the goal. The rule-based agent chooses the nearest object when multiple are available and puts objects in containers whenever possible.
>
> > Q5: Lines 127--130: Why is the reaching range provided for the Child and Wheelchair agents, but not the Bicycle and Frail agents?
> >
> - Bicycle and Frail agents have no constraints on reaching range, and their reaching range is the same as the helper with reaching range $[0, 3]$. Therefore, we do not express it specifically in our paper.
>
> > Q6: Line 121: 'successful rate': Do you mean 'success rate'"?
> >
> - Thank you for catching that error. We’ve revised the text to "success rate.”
>
> > Q7: Lines 121--122: What do you mean by 'the excess part'?
> >
> - The excess part refers to the amount a measurement exceeds a predefined limit. For example, if the reaching scope of an agent is $[0.25, 1.5]$ and the object height is $2$, the excess part is $2 - 1.5 = 0.5$.
>
> > Q8: Which of the actions in the action space are relevant to the emergency events? How does the helper agent stop the child from running away?
> >
> - In the emergency event scenario, the helper is ordered by the kid’s parent, the constrained agent, to stop the kid from running away too far, which can cause danger.
> - The environment is designed so that when the helper is near the child, the child **stops running away and returns to the constrained agent**, mimicking real-life interactions. There are no actions specifically designed for emergency events to ensure **the consistency of the benchmark**. Regarding the available plan for the decision module in the LM-based and rule-based helpers, one more plan, ‘*following [child agent],’* is available.
>
> > Q9: Table 1: 'Emergent event' -> Should this be 'Emergency event' instead?
> >
> - Thank you for pointing out this improper use of words. We agree with your assessment and have updated the term to "Emergency event.”
>
> > Q10: Line 298: 'Object spaceship' -> What does this mean?
> >
> - We apologize for this typo. It should be changed to object space information. This paragraph discusses the phenomenon that, given the locations of objects and the agent, LLM fails to infer which object is close to the agent.
>
> We hope our response has addressed your concerns and positively influenced your assessment. If you have further questions, please don't hesitate to reach out during the rebuttal period. Thank you for your valuable suggestions and comments—we truly appreciate them!

---

> > ### Author Response · Authors · 2024-08-21
> > **Thank you for your review and we are looking forward to your feedback!**
> >
> > Dear reviewer 8256,
> >
> > Thanks again for your suggestion to strengthen this work, we have revised the paper and conducted additional experiments to draw the 1-sigma error bar of the standard error in the PDF file attached to the global rebuttal following your suggestions. As the rebuttal period is ending soon, we wonder if our response answers your questions and addresses your concerns. If yes, would you kindly consider raising the score? Thanks again for your very constructive and insightful feedback!
> >
> > Best,
> >
> > Authors

---

> > > ### Author Response · Authors · 2024-08-25
> > > **Follow-up**
> > >
> > > Dear reviewer,
> > >
> > > Thanks again for your suggestion to strengthen this work. As the rebuttal period is ending soon, we wonder if our response answers your questions and addresses your concerns. If yes, would you kindly consider raising the score? Thanks again for your very constructive and insightful feedback
> > >
> > > Best,
> > >
> > > Authors

---

> > > > ### Author Response · Authors · 2024-08-28
> > > > **Follow-up**
> > > >
> > > > Dear reviewer,
> > > >
> > > > Thanks again for your suggestion to strengthen this work. As the rebuttal period is ending soon, we wonder if our response answers your questions and addresses your concerns. If yes, would you kindly consider raising the score? Thanks again for your very constructive and insightful feedback
> > > >
> > > > Best,
> > > >
> > > > Authors

---

> > > > > ### Comment · Reviewer_8256 · 2024-08-29
> > > > > **Response to rebuttal**
> > > > >
> > > > > Hi, apologies for the delay in responding, it's been a bit hectic preparing for the start of the semester. I appreciate the authors performing extra experiments to demonstrate the robustness of their results. I will increase my score from 5 to 6.

---

### Official Review · Reviewer_LsHw · 2024-07-25
**Constrained Human-AI Cooperation: An Inclusive Embodied Social Intelligence Challenge**

**Rating:** 8
**Confidence:** 4
**Correctness:** It appears so.
**Clarity:** The paper is well written.

**Review:**

The paper is clearly written and represents a valuable contribution to human-AI collaboration, however it could be improved by being clear about what the 'social' dimension of the 'intelligence' being explored within the paper actually is. The term 'social' is undertheorized and could be replaced with something like 'inter-agent' or 'contextual' intelligence without changing the meaning of the paper. This review would suggest that the term 'social' is far more capacious than the way it is being deployed here, and could be omitted or specified more carefully. Additionally, because the paper deals with user communities with accessibility issues, the paper could be improved by discussing what past, current, or future plans are for participatory engagement with such groups has been or might be.

**Strengths:**

The paper is appropriately scoped, makes clear contributions, and is well written.

**Additional Feedback:**

No additional feedback.

**Documentation:**

Yes.

**Limitations:**

Because the paper deals with user communities with accessibility issues, the paper could be improved by discussing what past, current, or future plans are for participatory engagement with such groups has been or might be.

**Opportunities For Improvement:**

The paper could be improved by being clear about what the 'social' dimension of the 'intelligence' being explored within the paper actually is. The term 'social' is undertheorized and could be replaced with something like 'inter-agent' or 'contextual' intelligence without changing the meaning of the paper. This review would suggest that the term 'social' is far more capacious than the way it is being deployed here, and could be omitted or specified more carefully. Additionally, because the paper deals with user communities with accessibility issues, the paper could be improved by discussing what past, current, or future plans are for participatory engagement with such groups has been or might be.

**Relation To Prior Work:**

This paper cites a range of relevant prior work.

**Summary And Contributions:**

This paper provides baseline models to evaluate human-ai collaboration for humans with constrained capabilities. The challenge design and model are well constructed and presented.

---

> ### Author Rebuttal · Authors · 2024-08-17
>
> We are deeply grateful for the positive feedback on our manuscript, and we carefully considered your suggestions for improving our paper.
>
> > Q1: The term 'social' is undertheorized and could be replaced with something like 'inter-agent' or 'contextual' intelligence without changing the meaning of the paper.
> >
> - Thank you for your insightful suggestion regarding the term ‘social.’ We agree that the term may be undertheorized. In our revision, we will replace it with a more proper one like ‘contextual.’
>
> > Q2: Because the paper deals with user communities with accessibility issues, the paper could be improved by discussing what past, current, or future plans are for participatory engagement with such groups has been or might be.
> >
> - Our design considers explicitly the various needs of the disability community. In the future, more situations and tasks will be added.
> - Since we believe the current helper could be improved, we release this benchmark first. After a few iterations of the helper's design, we plan to engage the disability community via **online recruitment in a user study** to evaluate the assistance provided by our AI assistant. Through this test, we can know (1) which method is **most popular among the disability community** and (2) whether the metrics we designed are **consistent with their feelings.**
> - Meanwhile, although the user study can only be carried out virtually, it paves the way for future real-world applications.
>
> We sincerely appreciate the high scores you have awarded our submission. Thank you for recognizing the efforts we have put into our work. We are glad to discuss further if you have any questions or suggestions.

---

### Author Rebuttal · Authors · 2024-08-17

We sincerely thank the reviewers for their thorough examination and insightful suggestions. We have addressed each reviewer's questions and suggestions under the corresponding sections. We welcome further discussion and questions.

In response to Reviewer 8256's suggestion, we conducted additional experiments to explicitly draw the 1-sigma error bar of the standard error in the attached PDF file. With additional runs, our main findings in the paper still hold.

We are pleased that the reviewers recognized our contributions, which we believe support the viability and potential impact of our work, including:

- Precisely defined tasks towards effective human-AI collaboration;
- A thorough evaluation of the performance of various agents;
- Potential for positive social impacts.

We hope our responses below convincingly address all reviewers’ concerns. Again, we thank all reviewers for their time and efforts!

---

### Author Response · Authors · 2024-08-21
**Thank you and we are looking forward to your post-rebuttal feedback!**

Dear AC and all reviewers:

Thanks again for all the insightful comments and advice, which helped us improve the paper's quality and clarity.

The discussion phase has been on for several days, and we are still awaiting the post-rebuttal responses.

We would love to convince you of the merits of the paper. Please do not hesitate to let us know if we can offer any additional experiments or clarification to make the paper better. We appreciate your comments and advice.

Best,

Author

---

### Decision · Program_Chairs · 2024-09-26

**Decision:**

Accept (Poster)

**Comment:**

This paper introduces a challenge problem in the form of 'constrained' human-agent cooperation, where the constraints are limitations on the part of the human teammate (e.g., wheelchair use, diminutive height, etc.). The benchmark involves long-horizon tasks in a 3D environment, and the utility of the benchmark is empirically evaluated by agents that attempt to plan task performance in these environments, taking social (or contextual) intelligence requirements into account by recognizing the 'lead' agent's goals and constraints from egocentric visual input. The paper addresses a question of significant importance and applicability, and represents a novel dataset for evaluating socially-aware intelligent agents assisting humans with physical constraints. Concerns remain about the terminology ('social' intelligence in particular is underdefined) and the details of the experiments performed. The paper would also benefit from a detailed copy-editing pass.

Strengths:
- The problem space being considered is of significant and increasing importance, as agent helpers are likely to initially be of most use to people with constraints. The work is clearly differentiated from related work in the area.
- The challenge problems proposed and the empirically validated benchmark for addressing them are a good fit for the D&B track.
- Multiple reviewers specifically appreciate the inclusion of outdoor tasks, which tend to be under-studied in work in this area.
- The set of empirical demonstrations of multiple approaches to different tasks, including LLM-based approaches, represents a meaningful contribution to the SoTA in its own right. The additional experiments performed to verify the robustness of these approaches is appreciated.

Weaknesses:
- Participatory design is not, generally speaking, something that should be added in after the technology has been developed. Although the proposed updates to the paper wrt. participatory design would be helpful, the authors should seriously consider how to bring the target populations into the work as soon as possible, including during testbed development.
- The paper needs a proofreading pass to be in line with expectations for NeurIPS.